# The Impact of Project Manager Soft Competences on Project Sustainability

**Jian Xue [1], Zeeshan Rasool [1],\*, Aqsa Gillani [2] and Ahmad Imran Khan [3]**

[1] School of Economics and Management, Shaanxi University of Science and Technology, Xi'an 710021, China; xuejian@sust.edu.cn

[2] Department of Management Sciences, Comsats University Islamabad, Islamabad Capital Territory 45550, Pakistan; aqsa6888@gmail.com

[3] Putra Business School, University of Putra, UPM, 43400 Serdang, Selangor, Malaysia; ahmadimran.phd_fin18@grad.putrabs.edu.my

\* Correspondence: zeeshan_rasool114@hotmail.com

**Abstract:** The current study suggests a different and innovative view by testing a unique combination of variables, which are unproven in a single model for the purpose of increasing the ratio of sustainable projects. The project manager can use the model to look their projects and can compose necessary changes for better outcomes. The study objects to postulate the competence breach of project managers with regard to sustainability, and to deliver direction that how to fulfill the research gap. The given work is centered on the result of project supervisor soft capabilities on project sustainability mediated by innovation. To achieve this aim, deductive approach was adopted. The sample size of the study was 242 respondents, and data were collected from software houses. The collected data were then analyzed by doing the structural equation modeling in PLS-SEM in order to examine the relationships. The outcomes demonstrate positive impact of project manager soft competences on project sustainability and mediating impact of innovation among the relationship of project manager soft competences and project sustainability. Innovation is directly linked to project sustainable development, and was accepted, which aligns with the previous studies. This research reflects the role of project manager soft competences on innovation and project sustainability. The study is unique in its scope and implications as the focus is upon empirical investigation of the project manager soft competences and project sustainability in the context of Pakistan.

**Keywords:** project manager; soft competences; innovation; project sustainability; competency; development

## 1. Introduction

"Projects are renowned for making up a crucial part of companies and society's sustainable growth" [1]. The sustainability theory has connected with the triple bottom line related to the basic economic, social, and foundation triple base line that needed a new project perspective [2,3]. The viability of the project faces the growth cycle (method to understand the productivity of the project) in developing countries, which is probably the most important obstacle [4]. Businesses around the globe have implemented sustainability policies, and thus the concept of sustainability has become a valid component of decision making, and has gained increased attention in management literature and media [5]. The uniqueness of each project plays a challenging role as project manager and requires a variety of skills from it. Numerous studies are being conducted to determine the most needed skill for a project manager. The labor market is evolving, and new gaps are emerging. Thus, it is inferred that the "project manager has a key role in the project and this role provides the ability to control many aspects

of the project" [6]. When projects are recognized as having a crucial role to play in organizations and society's sustainable development [2], the academic debate arose on the relationship between project management and sustainability. "Project managers and programmers are well placed to contribute to sustainable management activities". Moreover, Debby Goedknegt, A. J. Gilbert Silvius and Marc de Graaf also recognizes this crucial position of project manager, who argues that the project manager has "a lot" of impact on the implementation of sustainability concepts in the project [7,8]. The actual sustainability obligation (proper implementation of project sequence model) can vary from project to project, but the project manager should still have a definitive or powerful role (Silvius et al. 2019).

The skills are not new to the project management industry. There is a great need to develop a comprehensive project management framework, which includes the main sustainability competencies. Project manager's competencies are expected to be static and not dynamic. Acceptance of diverse project management skills has significant consequences for both (professional and academic) fields. Cohesive project teams should create a leadership and authority structure to integrate functional elements, skills and discipline into a cohesive whole, to achieve the goals of the project [9]. The project team's key role is 'not to do the job but to organize the decision process'. The project manager's role in recognizing sustainability includes technical competencies.

Innovation describes both a mechanism and an outcome. Project management has a special connection to handling innovation, so that most creative tasks are performed by project. Innovation project processes also involve a gated mechanism designed to promote creative projects, with increased rates of monitoring as funding rises over the lifecycle of the project [10]. The project develops a specific approach, from conception through to execution and eventual value realization. Innovation is critical for the economic health of the developed world; innovation can be credited to as much as half of all GDP (Gross Domestic Product) growth. A manager who wants to guide and direct the important processes of innovation needs to be or become an innovation manager with notable leadership skills [11–14]. Sustainable performance improvement, which can be gregarious to the community, also means the company has to innovate.

A previous study did not concentrate on project management skills effect on project sustainability by innovation in Pakistan's software industry. The present research, through innovation in Pakistan's software industry, will emphasize the effect of project manager soft competencies on project development.

### 1.1. Software Industry in Pakistan

In the previous four years, the software industry in Pakistan has shown positive trends in growth in 2016–2017. Pakistan's computer industry exports amounted to $3.3 billion, up to $6 billion in the coming years [11], and are projected to grow to $5 billion. Export-oriented, local software markets are dependent on imports for the latest technology and services. Officially, leading international software brands from countries such as Germany, Spain, the United States, the UK and China have formed an ongoing event in Pakistan.

Recently, the software industry has shown outstanding growth as a result of outstanding efforts and new motivational forces and incentives for IT (information technology), such as the Arfa Kareem Tower in Lahore, Pakistan. The IT sector has been supported by some big opportunities, such as 100% foreign ownership, 100% repatriation of capital and profits, absence of income tax on all IT exports until June 2019, and tax holidays on investment properties until June 2024. A previous study did not concentrate on the impact of project management skills on the sustainability of projects through innovation in Pakistan's software industry. The present study will concentrate on the impact of soft skills project manager on soft skills project manager, through innovation in software industry in Pakistan.

### 1.2. Problem Statement

Pakistan's information technology sector has grown rapidly in the last two decades, and the IT industry has turned out to be an aspiring industry for the younger generation. Many software projects initiated in the government sector have not been able to produce the desired results, and have not been completed in the estimated timeframe. Sustainability projects are the most important issue facing the development process in developing countries. Project managers who lack competence are a "common cause of project failure" [12]. There is a great need to "develop a framework for project management that incorporates key sustainability skills" [15].

Earlier studies have shown that "project management standards do not address the enabling role of projects in sustainability" [13]. The aim of the study is to identify the "capacity gap for sustainability for project managers and to provide guidance on how to close this gap" [16]. The current study suggests a different and innovative approach, by testing a unique combination of variables that are unproven in a single model for the purpose of increasing the ratio of sustainable projects. The project manager can use the model to look at their projects and can make the necessary changes to achieve better results.

### 1.3. Research Objectives

To examine the influence of project manager soft competences on project sustainability.

- To examine the impact of mediating variable innovation on the association of project manager soft competences and project sustainability.
- To inspect the role of project manager soft competencies on innovation.
- To examine the role of innovation on project sustainability.

### 1.4. Research Questions

This study aims to answer the questions mentioned below.

- Do project manager soft competences drive project sustainability?
- What is the impact of project manager soft competences on innovation?
- What is the impact of innovation on project sustainability?
- Does innovation act as a mediator between project manager soft competences and project sustainability?

### 1.5. Significance of the Study

Limited research has been done on the impact of project manager soft skills (PMSC) on project sustainability (PS) through innovation, not only in the software industry in Pakistan, but also in the global software industry. Distinctive from past studies, the current study focuses on the software industry in developing countries, i.e., Pakistan. This study is the first attempt to observe the soft skills of the project manager linked to sustainability through innovation.

The current study suggests a different and innovative approach, by testing a unique combination of variables that are unproven in a single model, in order to increase the sustainability of the project with regard to the industrial/practical contribution. The study will contribute to the presentation of important guidelines and references for sustainability and innovation thinking in IT projects, and to the analysis of project manager competencies and sustainability.

## 2. Literature Review

### 2.1. Project Manager Soft Competences

"Project manager is considered as one of the main persons who have a greater contribution in driving a project towards its successful achievement" [17]. The term competence assets of activities

included the skills, knowledge, abilities and personal characteristics that are necessary for successful role accomplishment [18]. The project management competency development (PMCD) framework explains a project manager's competency as the process by which the project manager endlessly applies his/her knowledge, skills and personal behaviors, with the purpose of delivering projects that will meet the requirements of different stakeholders [19,20].

For instance, firstly the skills and competencies are keys for active communication with team members and conflict resolution. Secondly, the knowledge competencies recognized as professional practice gap of the learner can be based on a range of needs. Thirdly, experience competencies mean that the project manager uses information from a variety of sources, including personal experience and his/her own observations, to identify options and solve problems. Competence has also been used as an umbrella term covering almost everything that might affect performance [21].

The unicity of each project requires a variety of skills and a difficult role for the project manager. Many studies are carried out to find a most desired skill for a project manager [22]. The require competences of the project manager are static instead of fluid. It has important implications for both professionals and scholars that project managers' competencies are complex [23].

In project management, project managers' competencies are found to be central to their success [24]. Additionally, Gilbert Silvius and Ron Schipper identified that "Project management competence retention (PMCR) is positively connected with the project success rate of an organization" [25]. Project manager skills are exceptional project changes, and these are critical for administration abilities [26]. According to [27], they present the various types of project managers and their linkage to project sustainability. Recently, IT labor market analysis (2014) revealed that IT project manager is a key role for global and national implementation. This is no shock to IT and project managers, who have worked constantly to find the right people to cut their most important projects and programed. The position of project manager is based on the possibility and primary responsibility of determining and affecting the degree of sustainability of projects [11]. It requires responsibility to determine which sustainability considerations are important for the specific situation. Therefore, these functions should be used to enhance sustainability, both in projects and within the organization.

### 2.2. Innovation

Innovation is together a procedure and a result. It includes "production or adoption, assimilation, and exploitation of value-added novelty" [28], to produce outcomes such as new or improved products and services, or to create or improve production or management methods in organizations [29]. The definition of innovation has been broadened beyond the outputs of research and development or technology, to include assets with a knowledge base, social and enterprise innovation and business models, as previously defined by [30].

### 2.3. Project Sustainability

Organizations are running with continuous pressure of changing conditions, especially in the areas of socio-economical, environmental and technological aspects, with the realization that now the accountability and organization performance are not being evaluated only in terms of financial audit and financial profitability [31]. Sustainable development is the development that meets the needs of the present, without compromising the ability to future generations to meet their own needs [22].

Project sustainability management defines "the discipline of planning, controlling, organizing resources, time and quality in order to successfully complete the project and classical matrices focuses to save resource, time and force. The classical method for project evaluation is emphasis in perspective of scale, time and money" [15]. Organizations achieve business objectives by undertaking projects, and the results of projects are considered for the assessment of business success. Sustainability has become a component of business success, and project management is one of the ways to get there. The content of sustainability, however, is addressed under the key word of "project context".

Moreover, M.A. Khan states that project sustainability is "the percentage of project initiated goods and services that are still being delivered and maintained after five years of termination of implementation of the project" and "the continuation of local action stimulated by the project and generation of successor services and initiatives as a result of project built initiatives" [32]. However, project sustainability should also be viewed in terms of time, change and wasted resource dimensions.

The role of project manager lies in the possibility and primary responsibility of determining and affecting the degree of sustainability of projects [33]. It requires responsibility to determine which sustainability considerations are relevant to the particular situation. These functions should therefore be used to enhance sustainability, both in projects and within the organization [34]. We believe, regardless of project and client, that project managers have the necessary skills to integrate sustainability into projects when the project has an internal or external focus. The project manager is responsible for raising sustainability concerns and customer sustainability solutions, and therefore needs to adopt a comprehensive project approach, combine external and external approaches. Furthermore, Gilbert Silvius and Melvin R. Weber et al., also notes that the promotion of sustainability decisions and actions is an individual responsibility, whether or not clients explicitly request sustainability inclusion in the project [33,34].

Table 1 indicates the major literature and variables findings.

**Table 1.** Definitions of project manager soft competences (PMSC), innovation and project sustainability (PS).

| Nature of variable | Variable Name | Author/Year | Findings |
|---|---|---|---|
| Independent | Project Manager soft competencies | (Gillard 2009) | "Personal competencies within the project management field have been defined as those behaviors, attitudes, and core personality characteristics that contribute to a person's ability to manage projects" |
| | | (Crawford 2009) | "A behavioral approach to competency includes knowledge, qualifications, skills, and personality characteristics" |
| Dependent | Project Sustainability | (Munck Galleli and Souza 2013) | "Sustainable development and the concept of sustainability, connected to the triple bottom line economic, environmental, and social that has required a new perspective for projects" |
| | | (Gann 2000) | "Sustainable development is the development that meets the needs of the present without compromising the ability to future generations to meet their own needs" |
| Mediator | Innovation | (Martens et al. 2017) | "Innovation process consists of negotiation and management skills, technology and business mindset, and project management ability. Combining technical skills with the ability to manage trade-offs between technical concerns and business imperatives". |

## 2.4. Research Gap

Organizations are more keen to incorporate sustainability in their operations today. Project management may contribute to the success of this process, but little guidance about how sustainable use is applied in specific projects is available. Project sustainability is a dynamic and systematic term, which requires sufficient project management expertise to take responsibility for sustainability. The more effective a tool or programmer is, the higher the implementation criteria.

The research aims at defining the sustainability skills gap between the project managers and offering feedback on how this gap can be closed [35]. The current research provides a new and groundbreaking view by evaluating a novel combination of variables that have not been proven to increase the ratio of sustainable projects in a single model.

*2.5. Theoretical Foundation*

We have adopted a general management contingency framework for our current research, but this also applies to project management. A survey on the categorization framework for the project is in line with the contingency selection method, underlining the principle that "organizational structure must contribute to the context" [15,21–23,25–34,36–45].

We suggested 'sustainability oriented project theory,' which would include a clearer view for the project managers and their partners of their responsibilities, incentives, relationships and processes, to be discussed by project leaders in helping to make communities more sustainable in the short- and long-term. This project is a profitable business company in rapidly evolving environments. To maintain the rules and regulations, to ensure its 'license to operate' and to enhance its competitive advantage, the team is responsible for companies, management and mobilization of partners, by addressing the economic, financial, social and temporal aspects of its industry.

The base theory for this current study was the contingency theory, as it describes all the including variables very well. The core theme behind all the phases is the result which, in the circumstance of this study, is sustainability. As this theory states, there is no more 'best approach' to establish an innovative environment, to lead a project or to achieve sustainability. Rather, the ideal strategy depends upon the internal and external environment and affects the project performance as well.

*2.6. Conceptual Framework*

In this study, we used three variables: project manager soft competencies (PMSC), innovation (INN) and project sustainability (PS), and examined the impact of PMSC on PS, as well as the influence of PMSC on project sustainability through innovation. The research model of this study is shown in Figure 1.

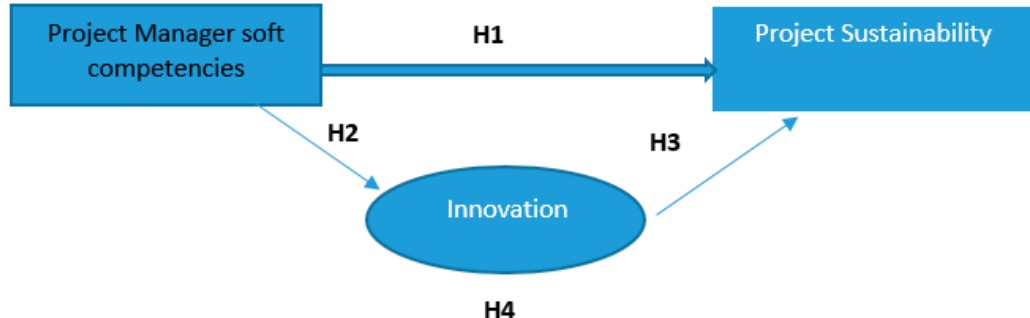

**Figure 1.** Conceptual Framework.

*2.7. Hypothesis Development*

**Hypothesis 1 (H1).** *Project manager soft competency has significant impact on project sustainability.*

**Hypothesis 2 (H2).** *Project manager soft competency has significant impact on innovation.*

**Hypothesis 3 (H3).** *Innovation has significant impact on project sustainability.*

**Hypothesis 4 (H4).** *Innovation significantly mediates the association between project manager soft competencies and, project sustainability.*

## 3. Research Methodology

### 3.1. Research Design

As indicated, the research involves significant and redefining the problems, generating hypothesis or suggested clarification. In simple words, research is the collection and evaluation of data. Research designs are those techniques and procedures for research that range the results from extensive range of assumption to itemized techniques for data collecting and analysis. The research relies on quantitative design, in which different techniques are used for data collection, such as surveys, questionnaires and rating scales. In this study, data has been collected through questionnaires. A questionnaire is an instrument comprised of a series of questions and different prompts to collect the data from respondents.

In this study, data analysis has been done by using PLS-SEM and SPSS software. Statistical package for the social sciences (SPSS) has been chosen to evaluate the data. Additionally, SPSS is an adaptable package that permits various kinds of analysis, data transformation and forms of outputs [36]. PLS-SEM has been used because researchers can execute realistic models, rather than developing multiple statistics and regression models. Using PLS-SEM, the researcher can estimate, determine and access their model as a graph, and characterize the relationship between the variables.

### 3.2. Sample Design

#### 3.2.1. Sample Selection

A convenience sampling method was used to collect the data. Convenience sampling is the type of non-probability sampling technique in which different units are chosen from the part of the population that is closed to hand. The sampling technique is used because it is speedy, readily available and cost effective. A sampling error can be ensued if the sample does not finish correctly reflecting the population it is supposed to signify.

#### 3.2.2. Population Frame

A population frame is the list or system that classifies members of the population so the sample can be drawn. It is basic of representative that is required to be perceived by the study. The population selected for this study is the software manufacturing of Islamabad and Rawalpindi, Pakistan. The entire quantity of recorded software communities in Pakistan is 1114. The total population in software houses of Islamabad and Rawalpindi is (2 * 325) 650. In this study, the total sample size was 242. The respondent is well educated and professional in their competency and they have at least a master's degree from well-known institutes. The data were collected from September to November in 2019.

Participants are key personnel on the projects that are project manager. They must have participated in leading the projects toward better sustainability. Data were collected from project managers. They must have knowledge about the learning process of the project. Thus, they know the best about the innovation and how it distresses the association amongst project manager soft competencies and project sustainability. Participants were provided the surety that their data would be remains confidential and only used for academic tenacity.

#### 3.2.3. Unit of Analysis

A project was taken as a unit of analysis. Thus, project managers were asked to respond in perspective of project sustainability, innovation and project manager soft competencies. These are the main research operational variables and taken information about listed variables through questionnaire (see Table 2).

**Table 2.** Variables and their measures.

| Sr. No | Name of Variable | Adopted From |
|--------|------------------|--------------|
| 1 | Project Manager Soft Competences | Rubin and Mason 1995; Gregory and Francis 2010. |
| 2 | Innovation | Vanhaverbeke et al. 2014; Daria et al. 2017 |
| 3 | Project Sustainability | Heepf 2008; Sherein 2014 |

### 3.2.4. Questionnaire

The questionnaire that has been used in this research consists of the following points. The first section was used for measuring project manager soft competencies, innovation and project sustainability. Questionnaire techniques are helpful for both (respondent and researcher) for the comprehensive understanding of research nature and feasibility. In that way, they can provide accurate and fruitful information about research variables and paradigms. The questionnaire is an effective tool to gather more information about variables because respondents are well educated and they can understand more, when all quires jour down in the form of a questionnaire. The sections are mentioned below,

- Project manager soft competences
- Innovation
- Project sustainability

### 3.2.5. Data Collection Procedure

This study collected and used primary data. Primary data were collected through personally self-administered questionnaires. For this study, the questionnaire technique of data collection has been chosen because questionnaires could be conducted simultaneously from a plethora of respondents. The self-administered questionnaire was distributed among the respondents of selected organizations of Islamabad and Rawalpindi, Pakistan. Moreover, data were collected from the project manager.

## 4. Data Analysis and Discussion

### 4.1. Normality Analysis

After the data recorded once, it is measured by using normality tests. To investigate normality within the variables, skewness and kurtosis tests are done [46]. The values of skewness and kurtosis must lie between +2 to −2 for the data are acceptable to prove the normal distribution [47] (Table 3).

PMSC, INN, and PS were distributed normally, and skewness calculated was good in the range −2 to s$^{+2}$ and this makes it good for hypothesis testing. Similarly, kurtosis fell well in this range, so the results showed the normality of the data.

### 4.2. Reliability

To test the coherence of the products used in the questionnaire, a reliability analysis is used. In order to calculate the reliability and the general use of Cronbach's alpha is to this effect.

The reliability of each variable examined is based on a selected sample. As can be seen from the table, the Cronbach's alpha value for the entire scale is above 0.7, demonstrating that the overall scale has good reliability. Independent variable project manager soft competencies having 10 items is 0.939, implying that this variable is reliable, as it falls within the range. The Cronbach's alpha of dependent variable project sustainability having 11 items is 0.944, implying that this variable is reliable, as it falls within the range. The Cronbach's alpha of mediating variable innovation having 11 items is 0.951. This implies that this variable is reliable. Thus, Table 4 shows that all the variables are reliable, with a high reliability level. "One possible reason for this high reliability is that the construct of social identification appears to be quite homogeneous". As noted, "scale has high inter-item correlations,

and the different components of self-investment in particular were all highly correlated both in the present studies" (Table 4).

**Table 3.** Normality Analysis.

|  | Skewness Statistics | Kurtosis Statistics |
|---|---|---|
| PMSC1 | −1.082 | 0.018 |
| PMSC2 | −1.284 | 0.766 |
| PMSC3 | −1.169 | 0.245 |
| PMSC4 | −1.056 | 0.027 |
| PMSC5 | −1.213 | 0.408 |
| PMSC6 | −1.252 | 0.206 |
| PMSC7 | −1.202 | 0.501 |
| PMSC8 | −1.204 | 0.587 |
| PMSC9 | −1.179 | 0.127 |
| PMSC10 | −1.024 | −0.088 |
| INN1 | −1.415 | 1.191 |
| INN2 | −1.487 | 1.322 |
| INN3 | −1.520 | 1.350 |
| INN4 | −1.350 | 1.099 |
| INN5 | −1.445 | 1.196 |
| INN6 | −1.493 | 1.378 |
| INN7 | −1.435 | 1.484 |
| INN8 | −1.361 | 1.097 |
| INN9 | −1.492 | 1.318 |
| INN10 | −1.415 | 1.379 |
| INN11 | −1.454 | 1.205 |
| PS1 | −1.411 | 1.319 |
| PS2 | −1.441 | 1.331 |
| PS3 | −1.461 | 1.204 |
| PS4 | −1.372 | 1.148 |
| PS5 | −1.446 | 1.352 |
| PS6 | −1.444 | 1.194 |
| PS7 | −1.423 | 1.344 |
| PS8 | −1.327 | 1.017 |
| PS9 | −1.402 | 1.196 |
| PS10 | −1.407 | 1.285 |
| PS11 | −1.419 | 1.266 |

**Table 4.** Reliability Analysis of Variables.

| Variable | Cronbach's Alpha | Number of Items |
|---|---|---|
| Project Manager Soft Competencies | 0.939 | 10 |
| Innovation | 0.951 | 11 |
| Project Sustainability | 0.944 | 11 |

### 4.3. Variance Inflation Factor Analysis

To check the multi-collinearity, we examined tolerance and the variance inflation factor (VIF), that is two collinearity analytical factors. They help to classify multi-collinearity. Values of VIF that exceed 10 are often regarded as demonstrating multi-collinearity, but in some models, values above 2.5 may be a cause for concern. Table 5 shows the values of tolerance and VIF.

**Table 5.** Variance Inflation Factor Analysis. Coefficients [a].

| Model | | Collinearity Statistics | |
|---|---|---|---|
| | | Tolerance | VIF |
| 1 | PMSC | 0.796 | 1.257 |
| | IN | 0.796 | 1.257 |

a = Dependent Variable: PS.

### 4.4. Data analysis in PLS-SEM

The following data techniques were run in the PLS-SEM:

1. Algorithm analysis
2. Bootstrapping analysis
3. Structural equation modeling

Algorithm Analysis

In order to evaluate the measurement model, CFA was conducted, and the results are shown below. Cronbach's alpha, CR estimates and AVE are higher than the cut-off values of 0.7 and 0.5, respectively (Table 6).

**Table 6.** Construct Reliability and Validity.

| | Cronbach's Alpha | Composite Reliability (CR) | Average Variance Extracted (AVE) |
|---|---|---|---|
| INN | 0.977 | 0.980 | 0.816 |
| PMSC | 0.941 | 0.950 | 0.654 |
| PS | 0.977 | 0.980 | 0.816 |

**Note**: PMSC = Project Manager Soft Competencies, INN = Innovation, PS = Project Sustainability.

"One possible reason for this high reliability is that the construct of social identification appears to be quite homogeneous. As noted, scale has high inter-item correlations, and the different components of self-investment in particular were all highly correlated both in the present studies and in [38]".

The discriminant validity tests measurements that are not supposed to be related are actually unrelated. Three methods are here for the evaluating discriminant validity of constructs, cross loadings, Fornell–Larkcer criterion (1971) and heterotrait monotrait ratio (HTMT). The heterotrait monotrait ratio (HTMT) has been used to evaluate the discriminant validity proposed by [48].

Value of HTMT should not exceed 0.90, as it can cause lack of discriminant validity in the constructs. Table 7 depicts that none of the value of HTMT is exceeding the benchmark of 0.90, so discriminant validity is established here.

**Table 7.** Discriminant Validity.

| | Innovation | PMSC | Project Sustainability |
|---|---|---|---|
| PMSC | 0.74 | | |
| Project Sustainability | 0.77 | 0.82 | |

The sample model fit is accessed by $R^2$ statistics. It is used to measure the predictive accuracy of the research model. $R^2$ indicates total variance of dependent variable explained by the independent variable. Benchmark for $R^2$ is 0.3, table depicts $R^2$ values of dependent variables are in acceptable range (Table 8).

**Table 8.** In-sample Model Fit.

| Varibales Name | R Square |
|---|---|
| Innovation | 0.866 |
| Project Sustainability | 0.773 |

Predictive power of the research model is measured by many researchers using $R^2$, but this is not the correct interpretation, because $R^2$ is the indicator of in-sample explanatory power of the model. There is a lack of out-of-sample predictive power of the model in R2 statistics. In addition, [14–19] suggested a set of techniques to address this concern. PLS prediction is used to measure the predictive power of model in PLS. It helps in quantifying with prediction error i-e root mean square error (RMSE) and mean absolute error (MAE). The main focus, while making interpretations of the outcomes of PLS prediction, should be on key dependent variable instead of endogenous variables (Table 9).

**Table 9.** Out of Sample Model Fit.

| MV Prediction Summary (PLS) | | | | MV Prediction Summary (LM) | | | Comparison (PLS−LM) | |
|---|---|---|---|---|---|---|---|---|
| | RMSE | MAE | $Q^2$_Predict | | RMSE | MAE | $Q^2$_Predict RMSE | MAE |
| IN9 | 0.406 | 0.264 | 0.799 | IN9 | 0.538 | 0.44 | 0.648 | −0.132 | −0.176 |
| IN8 | 0.472 | 0.356 | 0.811 | IN8 | 0.647 | 0.497 | 0.645 | −0.175 | −0.141 |
| IN1 | 0.487 | 0.348 | 0.834 | IN1 | 0.861 | 0.744 | 0.479 | −0.374 | −0.396 |
| IN7 | 0.38 | 0.263 | 0.772 | IN7 | 0.396 | 0.329 | 0.753 | −0.016 | −0.066 |
| IN11 | 0.445 | 0.315 | 0.789 | IN11 | 0.557 | 0.464 | 0.669 | −0.112 | −0.149 |
| IN3 | 0.429 | 0.313 | 0.822 | IN3 | 0.629 | 0.533 | 0.618 | −0.2 | −0.22 |
| IN2 | 0.381 | 0.245 | 0.845 | IN2 | 0.553 | 0.46 | 0.673 | −0.172 | −0.215 |
| IN4 | 0.525 | 0.369 | 0.849 | IN4 | 1.025 | 0.832 | 0.426 | −0.5 | −0.463 |
| IN5 | 0.5 | 0.386 | 0.772 | IN5 | 0.78 | 0.686 | 0.443 | −0.28 | −0.3 |
| IN10 | 0.481 | 0.341 | 0.742 | IN10 | 0.543 | 0.456 | 0.673 | −0.062 | −0.115 |
| IN6 | 0.408 | 0.288 | 0.806 | IN6 | 0.585 | 0.481 | 0.601 | −0.177 | −0.193 |
| PS2 | 0.438 | 0.311 | 0.757 | PS2 | 0.549 | 0.454 | 0.618 | −0.111 | −0.143 |
| PS11 | 0.447 | 0.317 | 0.754 | PS11 | 0.55 | 0.45 | 0.628 | −0.103 | −0.133 |
| PS8 | 0.515 | 0.383 | 0.781 | PS8 | 0.757 | 0.629 | 0.526 | −0.242 | −0.246 |
| PS5 | 0.381 | 0.26 | 0.803 | PS5 | 0.4 | 0.337 | 0.783 | −0.019 | −0.077 |
| PS6 | 0.429 | 0.32 | 0.801 | PS6 | 0.625 | 0.528 | 0.578 | −0.196 | −0.208 |
| PS1 | 0.433 | 0.285 | 0.837 | PS1 | 0.706 | 0.548 | 0.565 | −0.273 | −0.263 |
| PS4 | 0.502 | 0.365 | 0.786 | PS4 | 0.713 | 0.566 | 0.567 | −0.211 | −0.201 |
| PS3 | 0.437 | 0.32 | 0.773 | PS3 | 0.586 | 0.485 | 0.591 | −0.149 | −0.165 |
| PS9 | 0.415 | 0.301 | 0.796 | PS9 | 0.589 | 0.483 | 0.589 | −0.174 | −0.182 |
| PS7 | 0.412 | 0.297 | 0.762 | PS7 | 0.42 | 0.357 | 0.754 | −0.008 | −0.06 |
| PS10 | 0.455 | 0.297 | 0.821 | PS10 | 0.707 | 0.554 | 0.566 | −0.252 | −0.257 |

As the benchmark of factor loadings is 0.6, it can be seen from the Figure 2 that all the values of the items are higher than the standard range.

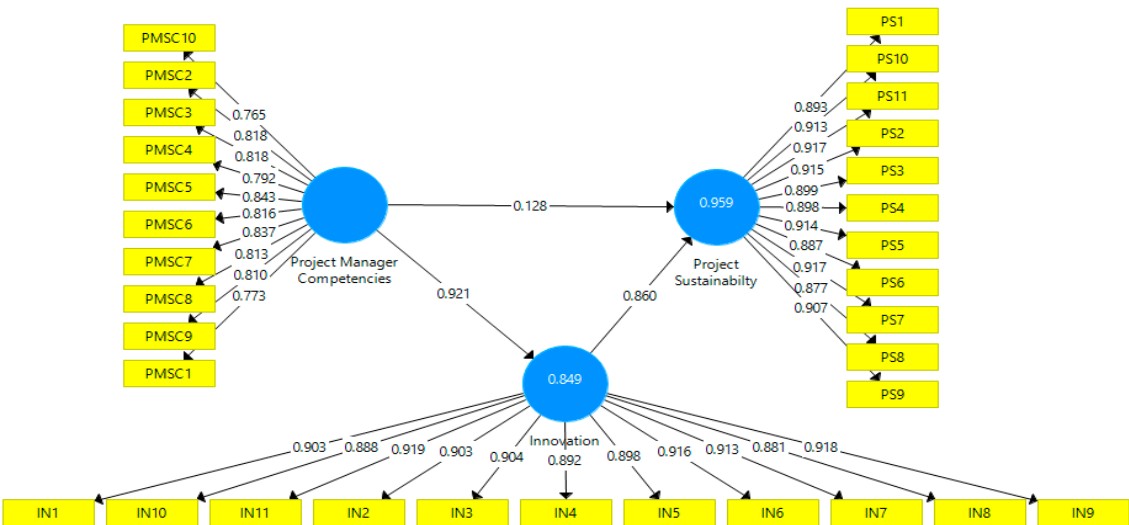

**Figure 2.** Factor loadings.

*4.5. Bootstrapping*

### 4.5.1. Path Coefficients

In order to make assessments of path coefficients, beta (β) values of all paths are being compared. Path coefficients are the relationships between the independent and dependent variables. The value of path coefficients should be more than 0.10, so they can show a specific effect in the research model suggested to test the significance of beta coefficients with the help of t-value, which is subsequently the result of a non-parametric technique bootstrapping.

Path coefficients can be seen in the Table 10. According to the results, we can say that all the paths have significant relationships. The T = 22.804 for innovation and firm project sustainability, which is within the standard range, and the same goes for the *p*-value, as *p* = 0.012. So, innovation has a significant effect on project sustainability. The T = 62.062 for project manager soft competencies and innovation, which is within the standard range, *p* = 0.000. So, entrepreneurial orientation has a significant effect on organizational learning. As the T = 3.357 for project manager soft competencies and project sustainability, which is within the standard range, and *p* = 0.001, as *p* = 0.001, which is less than 0.05. So, project manager soft competencies have a significant effect on project sustainability.

**Table 10.** Path Coefficients.

|  | **Mean** | **SD** | **T Statistics** | ***p* Values** |
|---|---|---|---|---|
| INN → PS | 0.859 | 0.038 | 22.804 | 0.012 |
| PMSC → INN | 0.921 | 0.015 | 62.062 | 0.000 |
| PMSC → PS | 0.129 | 0.038 | 3.357 | 0.001 |

**Note:** PMSC = Project Manager Soft Competencies, INN = Innovation, PS = Project Sustainability.

### 4.5.2. Specific Indirect Effects

In this study, the effect of project manager soft competencies on project sustainability mediated by innovation has been investigated. It indicates that there is not only mediation, but also a significant direct relationship between independent and dependent variables.

Table 11 mentioned shows the beta value (indirect effect), and *p*-value for mediation analysis. Moreover, β-value or indirect effect of project manager soft competences on project sustainability is 0.512. As the T = 6.521, and *p* = 0.000, it could be described that the mediation of the proposed

relationship is significant and this is because project manager soft competences have an impact on innovation and innovation has a significant impact on the project sustainability.

**Table 11.** Specific Indirect Effect.

|  | Sample Mean (M) | Standard Deviation (STDEV) | T Statistics (\|O/STDEV\|) | *p* Values |
|---|---|---|---|---|
| PMSC → Innovation → Project Sustainability | 0.512 | 0.079 | 6.521 | 0.000 |

*4.6. Hypothesis Summary*

4.6.1. Project Manager Soft Competencies Have Positive Impact on Project Sustainability

The results of data analysis show the beta coefficient value of project manager soft competences and project sustainability is 0.129. Hence, the first hypothesis, "project manager soft competencies has significant impact on the project sustainability". The analysis being done for this demonstrates that the project manager soft competencies are positive and significantly related to project sustainability (β = 0.129; t-value = 3.357, *p*< 0.001).

It also shows that if project manager soft competencies increase by 1 unit on average, then project sustainability will be increased by 0.129 units. This is in favor of our H1. Hence, null hypothesis for H1 is rejected, which is in line with the research question of the study that is what is the impact of the project manager soft competencies on project sustainability, and has achieved the objective stated to investigate the impact of project manager soft competencies on project sustainability.

4.6.2. Project Manager Soft Competencies Have Significant Impact on the Innovation

The results of data analysis show the beta coefficient value of project manager soft competences and innovation is 0.921. Hence, the second hypothesis, "project manager soft competencies has significant impact on the innovation". The analysis being done for this demonstrates that the project manager soft competencies is positive and significantly related to the innovation (β = 0.921; t-value = 62.062, *p* < 0.001).

It also indicates that if, project manager soft competencies increase by 1 unit on average, then innovation will be increased by 0.921 units. This is in favor of our $H_2$. Hence, the null hypothesis of $H_2$ was rejected, which is in line with the research question of the study; that is, what is the impact of the project manager soft competencies on innovation. This has achieved the objective stated to investigate the impact of project manager soft competencies on innovation.

4.6.3. Innovation Has Significant Impact on Project Sustainability

The results of data analysis show that the beta coefficient value of innovation and project sustainability is 0.859. Hence, the third hypothesis, "innovation has significant impact on project sustainability".

The results indicate that innovation has constructive and noteworthy effects on project sustainability (β = 0.859; t-value = 22.804, *p* < 0.001). This shows that if innovation increases by 1 unit on average, then project sustainability will be increased by 0.859 units. This is in favor of our $H_3$. Hence, the null hypothesis of $H_3$ is rejected. Additionally, this hypothesis is in line with the research question of the study; that is "what is the impact of innovation on project sustainability and has achieved the objective stated to examine the influence of innovation on project sustainability".

4.6.4. Innovation Significantly Mediates the Effect of Project Manager Soft Competencies on Project Sustainability

The results of data analysis show the beta coefficient value of innovation and project sustainability is 0.512. Hence, the fourth hypothesis stated that, "innovation significantly mediates the effect of project

manager soft competencies on project sustainability". Results suggest that the innovation significantly and positively mediates the effect of project manager soft competencies on project sustainability (β = 0.512; t-value = 6.521, *p* < 0.001). This shows that if project manager soft competencies increase by 1 unit on average, then project sustainability through innovation will be increased by 0.512 units. This is in favor of our H4. Therefore, the null hypothesis of H4 is rejected, which is in line with the research question of the study; that is, what is the impact of the project manager soft competences on project sustainability edited by innovation and has achieved the objective stated to investigate the impact of project manager soft competencies on project sustainability mediated by innovation.

## 5. Findings and Discussions

### 5.1. Conclusions

This study is done with the aim to examine the impact of project manager competences in achieving project sustainability; furthermore, it investigates the mediating role of innovation between project manager competences and project sustainability. To this extent, the data of 221 software houses have been collected, and quantitative analysis has been conducted by applying different analysis. The first two chapters include a broad review of literature used relevant to this study, a broad conceptual framework, rationale of the study, research gap, problem statement, research objective, significance of the study and structure of the thesis, and hypothesis. Chapter three explains how the appropriate methodology is formed for the study, how the data are collected and testing of the relationships of the hypothesis. Section four explains how the quantitative study is done. Descriptive analysis and hypothesis testing are done using sampling by PLS. The analysis shows that project manager soft competences are positively and significantly associated with project sustainability. Results also revealed that innovation mediates the relationship between PMSC and PS.

Industrialized and developing countries both face the process of development; this is the most major challenge in project sustainability. "Sustainability has become a component of business success, and project management is one of the ways to get there," [49]. In order to create a reasonable use of present resources and to offer a normal life for future generations, sustainability unites social, environmental, and economic responsibility [50]. "The actual responsibility for sustainability may differ by project, but the project manager always will have a decisive or influencing role [51]". Silvius suggests that project managers need a basic understanding of sustainability flexible definition for incorporating sustainability into projects and project management practices. However, project managers need to achieve a long-term and integrative view of the project, and that the project manager is completely responsible for the consequences of the project, including sustainability related outcomes. In order to better control organization and better customer relations in line with managing these trade-offs, the project management and role of project managers are currently receiving continuous attraction as project management by higher authorities of organization's; "project manager is responsible to integrate all the aspects of projects in line with the organizations strategy". Therefore, a study was required that could enlighten the importance of project manager soft competences and innovation for sustainable developments. A cross-sectional research was conducted from different software companies situated in Pakistan. Two types of relations were investigated here namely; the direct effect and the mediating effect. The mediating variable was innovation, whose mediation was inspected, between the relationship of project manager soft competencies and project sustainability. These days, projects are exceedingly focused and globally challenged, where a project manager plays its role as the main player. Project manager competencies brings innovation to meet with challenges, workers' mindset, and paving way to some new worldwide opportunities and help to improve project sustainability. So, this study highlighted the importance of project manager soft competencies, their influences for innovation and project sustainability. Reliability and validity analysis and mediation analysis were used for the analysis of data gathered through SPSS and PLS.

The first hypothesis is "Project Manager Soft Competencies has significant impact on the project sustainability". The analysis being done for this demonstrates and results show that the project manager soft competencies are positive and significantly related to the project sustainability. The second hypothesis is "Project manager soft competencies have significant impact on the Innovation". The analysis being done for this demonstrates results showing that the project manager soft competencies are positive and significantly related to the innovation. The results of data analysis show that the beta coefficient value of innovation and project sustainability is 0.859. The third hypothesis is "Innovation has significant impact on project sustainability". The results of data analysis show the beta coefficient value of innovation and project sustainability is positive. The fourth hypothesis stated that "innovation significantly mediates the effect of project manager soft competencies on project sustainability". The results suggest that the innovation significantly and positively mediates the relationship of project manager soft competencies on project sustainability. This study endeavored to push the limits of the present learning of project manager soft competences by investigating the wonder at the project level, characterizing it, and distinguishing its predecessors and esteem-based outcomes. The innovative ideas that the manager come up with can help the team to achieve their tasks and goals and ultimately complete their project with success that will lead them towards achieving sustainability in projects. Project manager matters a lot in this scenario, as it will boost the working side of the project and will bring innovativeness that will overcome difficulties in way of achieving sustainability. It additionally offers direction to project managers who are looking for approaches to manage expanding many-sided quality, and dangers, while keeping up long haul survival, versatility, and manageability of their project. A solid establishment for a progressing system of research is additionally settled.

### 5.2. Contribution

This study provides direction for project managers to identify required competences for a successful management of project, particularly by strengthening the environment for innovation. Maintaining a well-functioning system of project could help managers to create an effective environment for innovation, to meet the challenges of the external and internal environment, and avoid repeating the same mistakes. This study will contribute in presenting important guidelines and references for achieving sustainability and thinking about innovation in IT projects that are required for achieving sustainable IT projects, analyzing project manager competencies, and project sustainability. Project manager candidates can use this research to highlight their soft competencies in their resumes. A major contribution of this research gives good understanding of project competences, by explaining and defining the relations between project manager competences and project sustainability and their orientations. This research provides new important insights within the project by filling the existing literature. It also gives empirical evidence on the impact of project manager competences on project sustainability and its potential impact on innovation. In the previous literature, many unexplored constructs and interrelationships among them need clarification. The findings of this research suggest that more attention should be given to cultivating a specific set of competences for managers and an innovative environment, with the purpose of creating a sustainable project.

### 5.3. Limitations and Future Recommendations

There are certain limitations of this study. First of all, due to time constraint, this research is conducted only in the software industry; this is why it may or may not be generalizable to other industries, and it should be conducted in other industries to represent accurate results. Another limitation relates to a cross-sectional design of the study, in which the analysis is limited to one point in the time assessment. Due to small sample size, we could not use the structural equation modeling approach, because it does not meet the criteria; this is why the hypothetical model is not fully explained. Lastly, there might be some other factors which can affect the dependent variable. This research only took project manager soft competences and innovation into consideration. So, some other moderators or mediators should be included in research, and again measure the estimates, because supply chain

resilience is more important if, the logistics are to be successful. However, this study is a very important step towards project manager's soft competences with project sustainability in the software industry of Pakistan.

**Author Contributions:** This article is the result of the joint work by all authors who equally contributed to conceive, design, and carry out the research. All authors collaborated in analysing the data, preparing the data visualization and writing the paper. All authors have read and agreed to the published version of the manuscript.

**Funding:** This research is funded by "The natural science basic research plan in Shaanxi province of China" and project number is 2018JM7005.

**Conflicts of Interest:** The authors declare no conflict of interest.

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
