# Peer review of "The Impact of Project Manager Soft Competences on Project Sustainability"

_sustainability, doi:10.3390/su12166537_

Round 1

Reviewer 1 Report

I think that this work is ambitious, but the way in which it is presented needs very much work. In the following I will explain why it is not ready for me for publication.

I think that the section 2.1. Project Manager Soft Competences is made by many too short sentences. I would prefer a harmonized discussion. This problem is very present throughout the paper.

I really do not understand what you mean with sustainability, in particular in software development project. I think it is not acceptable to have only 6-7 lines of text to explain your meaning of sustainability, as well as it is not acceptable to see that you close the section 2.3 Project sustainability with: "The classical methods for measuring the performance of the project include scale, time and expense (Martens et al., 2017)". What do you want to highlight with such statement?

When you say "Sustainable development is the transition that meets the needs of today without losing their ability to fulfill their own needs for future generations" I would cite that is the Brutland definition, at least.

For section 5.3. Limitations and Future Recommendations, I suggest to include some other limitations (you say that they are more than one).

These are only some indication of rework needed, but the the problem is at the syntactic level. Most of the time the section are made by very short sentences without any connection with what is before and what is after that. The discussion is completely disconnetted in some parts. From this point of view the paper needs an important review by authors.

Line 164. Sustainability and sustainability are the three-pronged...

line 160. I would not start a sub section with "in particular, ..."

Line 40. It is not clear the reference and page number of the quote "Project managers and programmers are well placed to contribute to sustainable management activities".

Line 127. Skills is repeated twice. "the skills , knowledge, skills and personal characteristics"

Line 133 "second is repeated twice. "Second, knowledge skills known as a professional practice void for 133 the learner can be focused on a range of needs. Second, knowledge skills mean that the project 134 manager incorporates information from a variety of outlets,"

Line 138. something is missing. "The required competence of the project manager as static instead of fluid" ?

Author Response

Respected Editor,

Thank you for your worthy notes and error concern related research. Basically, your concern points highlight the basic drawbacks of the research. I have updated all concern points as per your valuable directions and guidelines. I hope that you will consider my reply. I have extensively redraw the literature and describe each corner of your questions. I have attached the word file for better and one by one discussion about your question. If, you have more query please do let me know.

Regards,

Reviewer 2 Report

In my opinion, the manuscript submitted for review should be rejected. Its main disadvantage is that it lacks determination of a cognitive gap and that the considerations presented therein are not based on the grounds of a specific theoretical approach.

Also, there are no relevant references to related and previous scientific works. The article does not contain an appropriate review of the subject-matter literature (e.g. Web of Science). The presented literature review is not comprehensive, complex, and logical. The results of other scientists who have so far dealt with the same problem have not been included therein.

There are no research hypotheses which should then be verified.

There is no description of the process of creating measures for measurement of variables (in the case of author’s measures); whereas, if these measures are authored by other scientists, there is a lack of relevant references to the literature in which these measures were previously described.

There are no connections between the obtained results and the findings of other scientists.
Such a drawback does not allow for identifying a contribution to the theory.
It is impossible to infer any relationships between variables from such simple descriptive statistics.  As a result, a cognitive value of the findings is very low.
No results of analysis of any relationships between the studied variables are presented. There are no clearly and properly described variables, as well.

Author Response

Respected Editor,

Thank you for your worthy note about my manuscript. Basically, I don't want to deliver any argument in favor of my manuscript. I heartily respect for your comments. 

Regards,

Reviewer 3 Report

It is an interesting topic that can contribute to the body of knowledge in the scientific arena. But I strongly believe that the authors should consider my following arguments in depth.

  1. All the paper needs extensive editing of English language for example: line 7-10 is not clear what the authors want to get across. Lines 29-30 what do they mean by "the growth cycle", line 43 "in or to the project" which is the difference?, line 43 "sustainability obligation" what do they mean?, lines 304-305 "are shown in below mentioned" wrong editing, Section 5.1 needs rewriting e.g. line 421 "it investigate..", lines 423-428 is not clear at all, line 430 "The analysis being done..., etc.
  2. Section 3 "Research Methodology" needs rewriting. Which is the questionnaire, what is the background of the respondents (experience, education etc.), when the research is conducted, there are variables as mentioned how these variables are analyzed and what was investigated. The forth variable is "Demographics", but in the context there is no reference what includes.
  3. In line 304 is mentioned the acronym CFA but there is no reference in the text what these letters stand for.
  4. The Section 5 "Findings and Discussions" is weak needs expansion for example: "Which are the project's managers characteristics that make them adequate to enhance projects sustainability?

I hope that this questions may enhance your work.

Author Response

(The authors gave the same response as above.)

Round 2

Reviewer 1 Report

I saw with pleasure that you have addressed all my comments and remarks. I appreciated the effort you put to improve the paper.

I don't know if it was wanted but lines 152-154 and 187 and 189 are repeated. Also, I would review the division in sections, I think that four levels are too much (es. 4.4.1.1.), maybe something can be merged in one section.

My last suggestion is to deeper analyze the literature about Project sustainability, also for future works. For example, I suggest a couple of references that could be useful to be included in the paper:

Marnewick, C., Silvius, G., & Schipper, R. (2019). Exploring patterns of sustainability stimuli of project managers. Sustainability11(18), 5016.

Armenia, S., Dangelico, R. M., Nonino, F., & Pompei, A. (2019). Sustainable project management: A conceptualization-oriented review and a framework proposal for future studies. Sustainability11(9), 2664.

Once integrated these two relevant references and revised the sections numbering, I think that your paper will be better.

Author Response

Respected Reviewer,

Thank you for your valuable comments and proper guidelines for manuscript corrections and shinning. Your way of highlights the errors basically boost my manuscript efficiency. I have added require material and quash mistakes as per your directions. If, you want to ask more query please do let me know.

Bundle of thanks for your time and dedications to evaluate my manuscripts and identification of errors and technical portions mistakes. 

With Profound Regards,

Reviewer 3 Report

Dear Authors,

Your manuscript was improved and you tried and answered most of my comments satisfactorily.

I still believe that your work needs proof editing some points do need correction, for example: Line 441 "further it investigate mediating"...

Also, the sections 3.2.3 & 3.2.4 need further development. There should be a short summary of the variables and what they include (the main research areas) and how these were investigated through the questionnaire.

With Regards.

Author Response

(The authors gave the same response as above.)
